# Research on the Potential Use of Grape Seed Flour in the Bakery Industry

**DOI:** 10.3390/foods11111589

**Published:** 2022-05-28

**Authors:** Oana Bianca Oprea, Mona Elena Popa, Livia Apostol, Liviu Gaceu

**Affiliations:** 1Faculty of Food and Tourism, Transilvania University of Brasov, 148 Castelului Street, 500014 Brasov, Romania; gaceul@unitbv.ro; 2Faculty of Biotechnology, University of Agronomic Sciences and Veterinary Medicine of Bucharest, 011464 Bucharest, Romania; 3National Research & Development Institute for Food Bioresources—IBA Bucharest, 6 Dinu Vintila Street, 021102 Bucharest, Romania; apostol_livia@yahoo.com; 4CSCBAS &CE-MONT Centre/INCE—Romanian Academy, Casa Academiei Române, Calea 13 Septembrie No. 13, 050711 Bucharest, Romania; 5Academy of Romanian Scientists, Ilfov Street, No. 3, 030167 Bucharest, Romania

**Keywords:** grape seed, bread, rheology, functional food, nutritional properties

## Abstract

Grape seeds are one of the most accessible by-products of the wine industry in large quantities (about 2.4 million t/year). Numerous researchers have shown that grape seeds have a high potential for use as a functional ingredient in the food industry due to their high content of protein, fiber, minerals, and polyphenols. The aim of the paper is to evaluate the possibilities of using grape seed flour (GSF) in the bakery industry from both chemical and rheological points of view. Research shows that grape seed flour contains about 42 times more fiber than wheat flour and approximately 9 times more calcium, 8 times more magnesium, and 2 times more potassium. To assess this potential, four samples of bread from flour mixtures with 3%, 5%, 7%, and 9% (*w*/*w*) degree of replacement with GSF were prepared, analyzed, and compared with a control sample from 100% wheat flour. From a rheological point of view, the baking qualities deteriorate: the water absorption capacity (CH) decreases from 58.2% to 55.8%, the dough stability increases from 8.50 min to 9.83 min, the α slope varies from −0.066 Nm/min to −0.104 Nm/min, the β slope increases from 0.576 Nm/min to 0.630 Nm/min, and the γ slope varies from −0.100 Nm/min to −0.198 Nm/min. The sensory analyses performed by the panel of evaluators enclosed the sensorial characteristics of the samples with 3% and 5% GSF between the two control samples made from flour types 480 and 1250. The conclusions show that the sample containing 7% and 9% are unsatisfactory from rheological and sensorial points of view and the samples with 3% and 5% can be considered a fiber source and a Cu source, respectively, and are rich in Zn.

## 1. Introduction

The common grape vine (*Vitis vinifera*) is a species of *Vitis* that is native to the Mediterranean region and today is cultivated almost on every continent [1]. In the wine industry, apart from the basic products, there are other products, known as by-products, representing 20–25% of the quantity of processed grapes [2]. Their capitalization is a matter of interest for specialists in the field because it leads to increasing the economic efficiency of the sector; offers the possibility of obtaining valuable products, useful in many industrial sectors, and the extraction of biologically active compounds (enzymes, vitamins, amino acids, etc.); and last but not least avoids environmental pollution.

According to a study conducted by Heuze and Tran in 2017 [3], within the program “Feedipedia” that is part of INRAE, CIRAD, AFZ, and FAO, seeds account for up to 6% of berry weight. The most important producers in the world are China (9.6 million t), the USA (6.6 million t), Italy (5.8 million t), France (5.3 million t), Spain (5.2 million t), Turkey (4.2 million t), Chile (3.2 million t), Argentina (2.8 million t), and Iran (2.1 million t). The EU accounts for 75% of grape production and 57% of wine production. The main wine producers are Italy, France, Spain (where most of the grape production is used for wine), and the USA. Assuming a worldwide production of 40 million t of fresh grapes used for wine, with a seed proportion of 1–6%, the potential amount of grape seeds can be estimated between 0.4 and 2.4 million t.

Grape seeds are by-products of wine-growing centers and the grape juice industry. These seeds contain lipids, proteins, carbohydrates, and 5–8% polyphenols, depending on the grape variety [4]. Grape seeds consist of dietary fiber; oils; proteins; phenolic compounds; and other important substances, such as minerals, vitamins, sugars, and organic acids. According to Mironeasa et al. [5], grape seeds have similar chemical composition to that of vegetables, being placed in the same category as pulse-protein products. Grape seeds contain some active compounds, such as dietary fiber, polyphenols, flavonols, and resveratrol; they are commonly used as a nutritional supplement [6,7,8].

Tita et al. [6] studied seven grape varieties and concluded that the phenolic compounds vary depending on the nature of the compound and the variety to which the seeds belong. For example, the varieties richest in syringic acid are Syrah and Novac (between 121.22 and 136.66 mg/L), followed by Burgund Mare and Cadarca. The lowest values are observed in the case of Cabernet Sauvignon and Merlot extracts. Gallic acid and vanillic acid are found most significantly in the Novac variety: 39.22 and 20.91 mg/L. Epicatechin gallate varies from 1.84 to 2.56 mg/L, the most significant values being in the case of seed extract from the Pinot noir variety. Caffeic acid reaches a maximum of 1.56 mg/L, specific to the Novac variety. Resveratrol is found in the seeds of all varieties, with values between 1.91 and 2.92 mg/L for Cabernet Sauvignon, 1.41 and 2.23 mg/L for Merlot, 1.93 and 2.37 mg/L for Pinot noir, 1.66 and 1.88 mg/L for Burgund Mare, 1.71 and 2.46 mg/L for Cadarca, 2.12 and 2.34 mg/L for Syrah, and 2.16 and 2.38 mg/L for Novac.

Grape epicarp (GE) and grape seeds (GS) have been used to produce flour (GEF/GSF), which can be used as a food ingredient in products such as bakery or pastry products, providing a novel method to solve the waste disposal issues besides the important benefits that these products bring to the consumers’ health [9].

In recent decades, consumer habits have undergone major changes, with preferences focusing on increasingly refined products, such as doughs with increasingly high baking properties but that are low in fiber and minerals [10]. The emergence of diseases related to these eating habits has led to the reorientation of the consumer to nutritionally balanced bakery products, the demand bringing to market nutritionally varied products obtained through innovative manufacturing technologies [11]. Grape marc, grape seed, epicarp, or other by-products left over from wine production contain many bioactive compounds that can be used in the bakery industry. These biocomposites are usually secondary metabolites whose characteristics vary depending on numerous pedoclimatic, technological, and biological factors [12].

Various pulverulent products rich in phenolic compounds, with important antioxidant properties and antimicrobial activities, have been obtained from tescovine by applying environmentally friendly processes that are economically accessible to the food industry. The obtained products have demonstrated high global antioxidant activities (ABTS dosage), having the role of delaying lipid oxidation and multiple antimicrobial properties [12,13,14].

The addition of grape seed flour to bakery doughs has been a topic of interest for many researchers. They tried to find out the optimal degree of replacement of wheat flour to preserve the rheological and enzymatic characteristics of the dough, as well as to achieve sensory characteristics that allow the acceptance of new products by the consumer [15,16].

With the increase in the amount of grape seed flour, two important phenomena are accentuated that require an in-depth study, namely:Lack of gluten and considerably high fiber content in grape seed flour, which lead to a decrease in the final volume of the bread and a decrease in the elasticity of the core;A bitter taste and persistence of aroma after chewing and swallowing, elements noticed in the sensory analysis.

Scientific research in the field of functional ingredients in the bakery industry has led to the production of products with varying degrees of replacement of wheat flour, for example, with partially defatted hemp seeds [17], Jerusalem artichoke tubers [18], oats [19], buckwheat [20], green tea [21], Pleurotus Ostreatus fibers (1,3/1,6 Beta-glucan) [22], various by-products from the fruit and vegetable processing industry, millets [23], sorghum [24], amaranth [25,26], quinoa [27], sunflower seeds [28], flaxseeds [29], pumpkin seeds [30], lupine [31], chia seeds [32], peas [33], chickpeas [34], lentils [35], chestnuts [36], and cricket flour [37], with important effects in terms of increasing the content of fiber, minerals, protein, vitamins, antioxidants, etc.

Laganà et al. [38] used bergamot *Pastazzo* flour (obtained after juice extraction), which is known for its high content of antioxidants, and obtained shortbread biscuits. *Pastazzo* flour (from pressed pulp) is usually used in animal feed, or it is discharged. The bergamot *Pastazzo* flour was used in different percentages (2.5%, 5%, 10%, and 15%), and the obtained results showed that the antioxidant content increases once the amount of *Pastazzo* flour added is increased.

Giuffrè et al. [39] fortified with olive oil an original recipe of Italian Cantuccini biscuits, using up to 70% extra virgin olive oil instead of 50% margarine and reducing by 20% the addition of cow butter. The aim of the study was to evaluate the shelf-life and physicochemical properties of biscuits and of the fats contained in original recipe Italian Cantuccini biscuits, and also the sensory properties were evaluated, including color, appearance, taste, flavor, texture, and overall acceptability.

Wandersleben et al. [40] conducted a study using three types of ingredients: lupine grit flour (AluProt-CGNA^®^, 60% of protein, d.m.), lupine hulls flour, and flaxseed expeller flour and tested the dough’s rheological properties with different combinations of the studied ingredients with wheat flour and also the consumers’ acceptance. The results show that the addition of natural sources of protein and dietary fiber through lupine or flaxseed flour to the wheat bread improved its nutritional profile. It was found to be the appropriate blend to attenuate the effect of foreign ingredients over the bread rheology, which normally interfere with the gluten network, reducing the quality of the bread, but also enhance its nutritional value.

Multescu et al. [41] studied the phenolic content, the flavonoid content, and the lipid-soluble antioxidant capacity of 14 by-products (rapeseed meals, grape seed flour, sunflower meals, seabuckthorn flour, etc.) obtained in the vegetable oil industry. Results confirmed that the by-products analyzed are a valuable source of many biological functional substances having considerable amounts of total phenolic content. The studied by-products can be used as ingredients for new bakery products in order to improve their nutritional properties and antioxidant quality.

Iriondo-DeHond et al. [42] studied the use of winery by-product extracts (grape pomace, seed, and skin) and a mixture of inulin-type fructans (inulin and FOS) as new ingredients for the development of yogurts with antioxidant and antidiabetic properties.

Libera et al. [43] used grape seed extract in the meat industry as a natural antioxidant, instead of sodium ascorbate, with optimistic results.

Amoah et al. [44] presented the possibilities of functional bread development through its valorization with certain plant-based by-products, such as grape pomace and grape seed extracts. An increased bioactivity of functional bread was observed by incorporation of flour from the plant-based by-products. In most cases, bread enriched with up to 6% flour from by-products had enhanced qualities. Regarding sensory acceptability of bread, formulations with up to an average of 5% flour from by-products resulted in bread with acceptable organoleptic perceptions of consumers.

Tremlova et al. [45] obtained “vegan” sausages with the addition of grape seed flour (GSF) in different concentrations (0%, 1%, 3%, 7%, 10%, and 20%). The results indicated that the GSF addition resulted in a higher antioxidant capacity of experimentally produced vegan sausages. Regarding the sensory evaluation, vegan sausages with 1% (according to taste evaluation) and 3% addition of GSF were the most acceptable by panelists. The sustainability of GSF usage is also achieved since it is a waste material generated worldwide within winemaking technology.

Král et al. [46] studied the addition of selected herbs and spices (ground cloves, cinnamon, mint, and grape flour) to biscuits, and the content of polyphenols and antioxidant capacity was measured, as well as their sensory properties and attractiveness to consumers. The results showed an increased antioxidant capacity for all samples, as well as for polyphenols. A 3% addition of the selected herbs and spices was confirmed to be overall acceptable to consumers. The conclusion based on the measurements is that a reasonable addition of natural substances containing natural antioxidants improves the overall quality of final products, in this case, biscuits.

Mironeasa et al. [4] studied the effect of grape seed flour (GSF) addition at the levels of 3, 5, and 7 g/100 g on the rheological behavior of the dough obtained from four types of wheat flour differentiated from a quality point of view. Iuga et al. [47] investigated the influence of the addition of different amounts of grape seed flour (from 3 to 9%) and particle size changes (large, L > 500 μm; medium, 200 μm > M < 500 μm; and small fractions, S < 200 μm) on the physicochemical characteristics of grape seed–wheat composite flours and the dough’s rheological behavior. Sporin et al. [48] used flour obtained from pomace of grape varieties (*Vitis vinifera*) Merlot (red) and Zelen (white), at 6%, 10%, and 15% substitution levels in wheat flour. The nutritional composition of 100 g of pomace flour from the two grape varieties (Merlot and Zelen) showed high levels of protein (11.3 g and 10.6 g, respectively), soluble fiber (3 g and 5.3 g, respectively), insoluble fiber (51.1 g and 44.3 g, respectively), lipids (13.9 g and 8.5 g, respectively), and carbohydrates (12.2 g, and 19.7 g, respectively).

The aim of this work is to evaluate the possibilities of using grape seed flour (GSF) in the bakery industry, as a functional ingredient, from both chemical and rheological points of view. In addition, the maximum replacement degree of wheat flour with GSF will be determined in the conditions of fulfilling the bakery standards and taking into account the sensory analyses of the panelists. Finally, nutrition labeling recommendations will be made based on the results of nutritional analyses.

## 2. Materials and Methods

### 2.1. Flour Mixtures

Raw materials used to prepare flour mixtures: wheat flour (WF) type 480 (ash = 0.48%) provided by M.P. Băneasa-Moară S.A., (Ilfov, România); mechanically defatted grape seed flour (GSF), from Romanian fruits, supplier SC 2Eprod SRL, (Teleorman, Romania).

In Table 1 are presented the five samples of mixtures of wheat flour and different proportions of grape seed flour. Comparative rheological analyses were carried out between the mixtures of GSF and 480 type wheat flour, and all four mixture samples were also compared to a control sample, namely to P0: 100% white wheat flour type 480.

### 2.2. Chemical Analysis

The moisture content of the flour mixtures was determined in compliance with ICC Standard No. 110/1 [49]. The ash content was determined by incineration at 525 ± 25 °C (ICC No. 104/1) [50]. Total nitrogen (N) and crude protein content (N·6.25, conversion factor) was determined by the Macro Kjeldahl Method (SR EN ISO 20483:2014) [51]. Total fat content was determined by extracting with petroleum ether at 40–65 °C, in compliance with the Romanian standard SR 91/2007 [52].

### 2.3. Crude Fiber Content Analysis

The content of crude fiber (cellulose, hemicellulose, and lignin) of the samples presented in Table 1 was determined by using Fibretherm–Gerhardt equipment. The sample is treated with an acid detergent solution (20 g *N*-cetyl-N,N,N-trimethylammonium bromide dissolved in 1 L sulfuric acid 0.5 M). Cellulose, hemicellulose, and lignin from the analyzed samples are insoluble in this solution, unlike other components. The dilution and filtration steps are simplified by using special fiber bags. The most important aspect of this method is adherence to strict boiling times and to weighing procedures. After treating the material with the acid detergent solution, the resulting insoluble residue is dried and weighted and afterward calcinated. The acid detergent fiber (ADF) content represents the insoluble part of the sample that is left after boiling in acid detergent solution, from which the ash obtained upon calcination is subtracted:(1)$$\%\;ADF=\frac{\left(\left(\chi -\alpha \right)-\left(\delta -\xi \right)\right)\times 100}{\beta }$$
(2)$$Blank\;value\;\left(\xi \right)=\gamma -\psi$$
where:*α* = mass of the fiber bag (g);*β* = sample mass (g);*χ* = mass of the crucible and the dried fiber bag after digestion (g);*δ* = mass of the crucible and ash (g);*ζ* = blank value of the empty fiber bag (g);*γ* = mass of the crucible and ash of the empty fiber bag (g);ψ = mass of the crucible (g).

### 2.4. Mineral Content Analysis

The mineral content was determined using an atomic absorption spectrophotometer (ContrAA 700; Analitykjena) [53]. Total ash was determined by incineration at 525 ± 25 °C in an oven. Analysis was performed using an external standard (Merck, multi element standard solution), and calibration curves for all minerals were obtained using 6 different concentrations. Dried samples were digested in concentrated HCl.

### 2.5. Rheological Properties Testing

The rheological behavior of the dough was determined using the predefined “Chopin +” protocol on Mixolab, according to ICC No. 173 [54], a protocol for complete characterization of flours (water absorption, protein quality, amylase activity, and starch quality), and a simplified graphic interpretation of the results was performed (Mixolab device-Chopin, Tripette et Renaud, Paris, France) [54].

The rheological behavior analysis of the Mixolab procedure parameters were the following: tank temperature 30 °C, mixing speed 80 rpm, heating rate 2 °C/min, and total analysis time 45 min. Mixolab curves recorded are basically characterized by torque in five defined points (C1-C5; N·m), with temperatures and processing times corresponding to them.

The relation of these features to the physical state of tested dough during mixing and heating [54] is:C1: Represents the maximum torque during mixing (used to determine water absorption) (Nm);C2: Measures the weakening of the protein chain by mechanical and thermal action (Nm);C3: Measures starch gelatinization (Nm);C4: Measures the stability of the starch gel (Nm);C5: Measures the degradation of starch in the cooling phase (Nm);TC1: Represents the duration of dough development (min);TC2: Represents the duration of weakening of the protein chain under mechanical and thermal action (min);TC3: Indicates the gelling time of starch (min);TC4: Indicates the stability time of starch gel (min);TC5: Indicates the duration of starch downgrading in the cooling phase (min).C1-C2: Shows the protein network strength on increasing heat;C3-C4: Denotes to starch gelatinization rate;C4-C5: Relates to the anti-staling effects (starch retrogradation in the cooling phase); represents the shelf life of the final product;α: Represents the strength of the protein chain (Nm/min);β: Represents the starch gelatinization rate (Nm/min);γ: Indicates the enzymatic downgrade speed (Nm/min).

The slopes of the curves C1-C2, C2-C3, and C3-C4 are represented by the coefficients α, β, and γ, which are calculated by Mixolab equipment.

The correlation between parameters (Table 2) is tested during dough mixing and heating processes by Mixolab.

### 2.6. Bread Making

Based on several preliminary tests performed on the bakery technology line of the Transilvania University of Brașov and consultations with specialists in the field, the technology for the manufacture of bakery products by the direct method was established. The manufacturing recipe used in the experimental research for obtaining bread samples by the direct method was: 1.5 kg of flour, 30 g of baking yeast, 15 g of salt, and 1050 mL of water; kneading for 7 min slowly and 5 min quickly in the mixer (Silver 50, Sigma, Italy); fermentation for 45 min in the leavener (Telbo, NovaPan, Romania) in a controlled atmosphere (32 °C; relative humidity 70%); portioning into 850 g pieces; round format modeling (T1, Gerosa, Italy); intermediate fermentation for 15 min; final modeling, final fermentation for 45 min in a fermenter under controlled temperature and humidity (32 °C; relative humidity 70%); baking in the oven (ring-type steam oven model MSR 4, NovaPan, Romania) for 35 min at a temperature of 230 °C, with steam treatment in the first 10 s; cooling to room temperature (minimum 2 h). Table 3 summarizes the recipe used to obtain the control sample P0 by the direct method. Samples P1, P2, P3, and P4 were obtained by replacing in the recipe P0 the amount of wheat flour indicated with a mixture of WF and GSF, as shown in Table 1.

### 2.7. Physicochemical Characteristics of the Experimental Bread

The bread’s specific volume was determined using the rape seed displacement method according to SR 91:2007, AACC 2000 [52,56]. The ratio of the obtained data was the average of triplicate measurements of the fresh made bread loaf, expressed in cm^3^/g. For porosity measurement, knowing the mass and density, the porosity was expressed in % by measuring the total scale of holes in a known crumb volume.

Elasticity content measurement consists in pressing a piece of bread crumb of a certain shape for a given time and measuring the return to the initial position/shape after removing the pressing force. Crumb elasticity is expressed in percent, meaning the ratio between the height expressed in % by pressing and return and the initial height of the cylinder crumb bread.

Moisture content measurement consists in drying approximately 5 g of bread crumb at 103 °C (±2 °C) to a constant weight; reported data consist in the mean of three measurements, each time performed on a fresh new bread loaf.

Acidity measurement, expressed in degrees [52,56], was determined by titration of a fluid extract of bread with 0.1 N NaOH solution in the presence of phenolphthalein as the indicator.

### 2.8. Sensory Analysis

A group of 10 specially trained panelist, with ages between 25 and 60, evaluated the bread samples, giving grades from 1 (lowest intensity) to 5 (highest intensity) for the following sensory attributions: crust color (degree of perceived brown color characterizing the crust), crumb color (degree of color darkness in the crumb, ranging from white to dark brown), crumb pore uniformity (size of pores on the surface (small/big), crumb softness (minimum force necessary to compress the sample), bitter taste (perceived by the back of the tongue and characterized by solutions of quinine, caffeine, and other alkaloids; usually caused by over-roasting), salty taste (fundamental taste sensation elicited by sodium chloride), sour taste (fundamental taste sensation evoked by acids, e.g., tartaric acid), specific aroma (aroma of fresh baked bread and odor associated with aromatic exchange from yeast fermentation), and after-taste (flavor staying after tasting). There was also a consumer overall acceptability determination on a 9-point hedonic scale (from 9 = I like it extremely to 1 = I dislike it extremely), where 35 untrained panelists with ages between 21 and 60 (70% females and 30% males) tasted the samples that were coded with 3 random letters in order to not influence their perception, and the results were expressed as a mean. The main conditions in choosing the panelists were for them to not be smokers and to have good health [57,58,59].

All bread samples of flour mixtures P1–P4 were compared to the standards of 2 control samples, P0 (wheat flour type 480; white bread) and PN (wheat flour type 1250; black bread).

### 2.9. Statistical Analysis

All analyses were executed in triplicate, and the mean values with the standard deviations were related. For statistical analysis was used the Microsoft Excel program, with the level of significance set at 95%. Analysis of variance (ANOVA) and Tukey’s test were used to estimated statistical differences between samples. Differences were considered significant for a value at *p* < 0.05.

## 3. Results

### 3.1. Chemical Composition

The chemical characterization of the grape seed and wheat flours used in this study is presented in Table 4. As can be seen, the nutritional composition of GSF shows the important potential for bakery product fortification through a high content of fiber and minerals. It is easy to notice that except for the smaller iron and zinc levels, the protein, fiber, potassium, magnesium, and calcium contents of grape seed flour are significantly higher than of wheat flour.

Comparing the data in Table 4 for crude protein, crude fat, total sugar, and total fiber, it can be seen that GSF has a crude protein content of 16.32% d.m., a value 35% higher than that of wheat flour (12.01% d.m.). The fat content of GSF is 5.92% d.m., which is more than 5.74 times the value of 1.03% d.m. determined for wheat flour. The total sugar content in the case of GSF is 11.31% d.m., 11.08 times higher than the value of 1.02% d.m. recorded in the case of wheat flour. Regarding the total fiber content, in the case of GSF, a value of 83.01% d.m. was obtained, a value 41.9 times higher than that of wheat flour (1.98% d.m.). From the analysis of the data in Table 4 for Ca, Mg, and K content, it can be seen that the GSF has a Ca content of 405.83 mg/100 g, a value 9.26 times higher than that for wheat flour (43.81 mg/100 g). The Mg content for GSF is 397.81 mg/100 g, 8.33 times higher than the value of 47.73 mg/100 g determined for wheat flour. The K content for GSF is 360.23 mg/100 g, a value 1.92 times higher than the value determined for wheat flour (187.13 mg/100 g).

The results of the performed analyses show that GSF contains a high amount of minerals. In addition, 100 g of grape flour assures a significant part of the daily intake of Ca, Mg, and K according to the Reference Daily Intake (RDI) of macronutrients and micronutrients recommended by the FDA and presented in Table 5. It can be seen that the content of potassium in GSF is almost double than that in wheat flour. The magnesium content level is 8 times higher for GSF, and the calcium content is almost 10 times higher in GSF than in wheat flour.

In Table 6 are presented the experimental results of the nutritional composition and mineral content of wheat flour mixtures with the addition of GSF. The addition of GSF resulted in significant increases in all four samples of mixtures in terms of fiber content, as follows: For sample P1, a relative increase of 125.75% was identified. For sample P2, the relative increase was 204.54%. Sample P3 registered a relative increase of 286.36%, and in the case of the last sample, P4, the relative increase was 368.18%. At the same time, there were significant increases in sugar content, 91.17% for the P4 sample (addition of 9% GSF); an increase in the fat content of more than 50%; and a slight increase (3.83%) in the protein content, proportional to the increase in the GSF addition.

Regarding the content of mineral substances for each of the studied samples P0–P4, the following are presented.

Significant increases were recorded in the Ca content, where an increase was observed from the value of 43.81 mg/100 g (sample P0) to the value of 76.40 mg/100 g (sample P4), an increase of 74.38%. In addition, there were increases of over 60% in the case of Mg content and 38.88% for Cu content. Increases below 10% were recorded for the K content. As for the Fe content, it decreased by 3.6% (P0: 1.11 mg Fe/100 g; P4: 1.07 mg Fe/100 g), and the Zn content decreased by 7.73% (P0: 5.43 mg Zn/100 g; P4: 5.01 mg Zn/100 g).

### 3.2. Rheological Properties of Flour Mixtures

In Table 7 and Table 8, the following rheological parameters determined using Mixolab equipment (ICC Standard 173) [54] Chopin + protocol for samples P0, P1, P2, P3, and P4 are presented: water absorption (%), CH, stability (min), ST, amplitude (Nm), α (Nm/min), β (Nm/min), γ (Nm/min), and C1, C2–C5 and TC1, TC2–TC5. Table 7 presents the rheological properties of wheat and GSF mixtures versus the P0 control sample of wheat flour.

The determinations performed for sample P0 (wheat flour type 480) showed rheological behavior specific to baking flour with average technological characteristics:

Water absorption (CH) = 58.2%; stability (ST) = 8.5 (min); amplitude = 0.1 (Nm); and TC1 dough development time = 1.2 min, similar results being obtained by Mironeasa et al. [15] and by Apostol et al. [61], which for water absorption (CH) recorded values of 61%, for stability (ST) obtained values of 8.9 min, and for TC1 presented values of 1.68 min.

Regarding the enzymatic activity of the flour in sample P0, characterized by the parameters α = −0.066 (Nm/min) (strength of the protein chain); β = 0.576 (Nm/min) (starch gelatinization rate); γ = −0.1 (Nm/min) (enzymatic downgrade rate); C2 = 0.461 (Nm), TC2 = 16.62 min; C3 = 2.041 (Nm), TC3 = 24.75 min; C4 = 1.553 (Nm), TC4 = 30.95 min; and C5 = 3.131 (Nm), TC5 = 45.02 min, this can be considered normal for average flours in the bakery industry. Baranzelli et al. [62] obtained similar values for wheat flour, namely: water absorption (CH) = 60.5%, stability (ST) = 11.34 min, C2 = 0.62 (Nm), C3 = 2.01 (Nm), C4 = 1.76 (Nm), and C5 = 3.24 (Nm). Numerous other researchers [62,63] have obtained similar results using the same analysis equipment and raw materials.

Comparatively analyzing the values determined for water absorption in the case of the five samples P0–P4 from Table 4 shows that as the percentage of grape seed flour increased, the water absorption capacity (CH) decreased, i.e., 58.2% (P0), 58.0% (P1), 56.7% (P2), 56.2% (P3), and 55.8% (P4), a maximum decrease from P0 to P4 of 4.1%, a phenomenon that can be explained by the lipid content of grape seed flour, which has a hydrophobicizing effect on the particles.

It is also noted that the minimum value reached in sample P4 is located at the lower limit of the optimal range of 55–62% for the manufacture of bakery products [64], which confirms the correct choice of the maximum level of flour grape seed in the case of sample P4. Sporin et al. [48], Mildner-Szkudlarz et al. [65], Samohvalova et al. [66], and Kuchtova et al. [67] obtained similar results of decreased water absorption.

The stability of the dough increased from 8.50 min (control sample P0) to 9.10 min (P1), 9.26 min (P2), 9.48 min (P3), and 9.83 min (P4), a maximum increase from P0 to P4 of 15.64% (Table 4). In terms of amplitude, there were no statistically significant changes. Similar results have been obtained by other researchers, such as: Mironeasa et al. [15], Sporin et al. [48], Kuchtova et al. [67], Aghamirzaei et al. [68], and Valková et al. [69]. Analyzing the rheological and enzymatic phenomena that take place after the period of stability of the dough, notable differences can be highlighted (Table 7 and Table 8) caused by the addition of GSF.

The legend regarding the definition of the coefficients (C1–C5 and TC1–TC5) has been presented in Section 2.6.

The values of the α slope, which express the strength of the protein chain, increased (in absolute value) from 0.066 Nm/min (P0) to 0.074 Nm/min (P1), 0.086 Nm/min (P2), 0.094 Nm/min (P3), and 0.104 Nm/min (P4), a maximum increase from P0 to P4 of 57%, (Table 8), which indicates a more intense degradation of the protein chains due to the mechanical and thermal action with the increase in the content of grape seed flour. Singh et al. [70] obtained similar results, with α values for 15 wheat samples ranging from 0.06 Nm/min to 0.1 Nm/min.

The values of the β slope, which express the gelling phenomena of starch, increased from 0.576 Nm/min (P0) to 0.588 Nm/min (P1), 0.604 Nm/min (P2), 0.618 Nm/min (P3), and 0.630 Nm/min (P4), a maximum increase from P0 to P4 of 9.3% (Table 8). The dough also reached a significantly higher consistency, C3 = 2.211 Nm (P4), compared to C3 = 2.041 Nm (P0), an increase (P0–P4) of 8.3% (Table 8). Analyzing the evolution of TC3 values in Table 8, it is found that the maximum consistency point C3 was also reached faster, indicating a significantly faster gelling process (a phenomenon also indicated by the increase in β slope values), visible in Figure 1. Singh et al. [70] obtained similar results, with β values for 15 wheat samples ranging from 0.06 Nm/min to 0.63 Nm/min.

γ slope values, which express the phenomena of amylolysis of starch gel, increased (in absolute value) from 0.1 Nm/min (P0) to 0.118 Nm/min (P1), 0.136 Nm/min (P2), 0.164 Nm/min (P3), and 0.198 Nm/min (P4), a maximum increase from P0 to P4 of 98% (Table 8), which indicates an intensification of the enzymatic activity. Analyzing the evolution of TC4 values in Table 8, it is found that point C4 was also reached faster, indicating a significantly more intense amylolysis process (a phenomenon visible in Figure 1). Singh et al. [70] obtained similar results, with values of γ for 15 wheat samples ranging from 0.003 Nm/min to 0.47 Nm/min.

### 3.3. Bread Quality

The analysis methods of the quality indicators for the bread samples subjected to the experimental research comply with the SR 91/2007 [52] standard and aim at determining the physicochemical indicators necessary for the evaluation of the bread quality. These indicators are: bread volume, bread core porosity, core elasticity, and bread acidity.

Four bread samples were obtained from the mixtures of wheat flour and GSF and were analyzed compared to the control sample P0. The coding of the experimental bread samples was identical to the coding of the flour mixtures in Table 1.

In Table 9 are presented the physicochemical indicators for all the studied samples, with values that are within the limits of SR 878/1996 [71], minimum 66%, in the case of porosity, and maximum 3.3%, in the case of acidity [71].

The volume of sample P1 (278 cm^3^) was within the permissible volume limits of white bread, but in the case of sample P2 (220 cm^3^), the volume was not even in the permissible limits for black bread (min. 225 cm^3^), according to SR 878/1996 [71]. It can be seen that the P4 sample volume, with the addition of 9% GSF, decreased to 197 cm^3^, compared to the volume of the P0 control sample, which was 290 cm^3^.

The porosity values of the sample with the lowest percentage of added GSF (3%) was 82% (P1), a value that is within the limits for white bread (min. 75%). As for the P2 sample, it did not reach the minimum limit of black bread (min. 62%, according to SR 878/1996) [71]. It is observed that a high percentage of GSF has visible negative consequences for the porosity of the bread (Table 9).

The elasticity of the P1 and P2 samples was within the limits allowed for a semi-white bread and that of the other samples within the limits for black bread.

Regarding the acidity of the bread samples, it can be seen that with the increase in the percentage of GSF, the acidity also increased to 2.2 degrees (P4), a value that is within the normal limits for white wheat bread (max. 3.3%) (SR 878/1996) [71].

The results of the analyses obtained concerning the evaluation of the physicochemical characteristics of the bread samples with added GSF confirmed the results obtained from the rheological and enzymatic analysis of the samples of flour mixtures. Thus, sample P1, the mixture of 97% wheat flour + 3% GSF, shows acceptable changes in the values of the rheological parameters for a good technological behavior and an acceptable quality of bakery products. In the case of the other samples, a negative influence of the rheological behavior is observed.

Figure 2 shows images of optically scanned bread slices for samples P0–P4. It is possible to observe the change in the color of the core and the crust, as well as the reduction in the porosity of the volume of the samples with the increase in the amount of GSF added.

### 3.4. Sensory Evaluation

The bread samples were subjected to sensory analysis 1 day after processing so that they were in similar conditions of consumption as the usual bread assortments in the bakery sales network. Experimental samples of bread with the addition of GSF were compared to both white bread (P0) and black bread (PN) and to each other to determine the optimal percentage of acceptable ingredients from the sensorial point of view in order to obtain a product with functional potential. For the 10 characteristics established in the sensory evaluation of the bread samples, equal importance factors were established, and in the interpretation of the results, the arithmetic mean of the scores received for each sample was used. This simplified calculation method was chosen as it is a comparative analysis in the process of developing a new bakery product. In Table 10, the sensory attributes and the score for each attribute used in the sensory assessment sheet are described.

As can be seen in Table 10, the color of the core of the samples darkened as the percentage of added GSF increased, the average of the marks given by the evaluators increasing from 1.17 (sample P0) to 4.81 (P4). Regarding the crumbliness of the core, the average of the marks given by the evaluators was 2.16 in the case of P0 and increased to a maximum of 4.63 in the case of P4 sample, which leads to the conclusion that with the increase in the amount of GSF, the quality of the samples worsened.

In terms of aroma and taste characteristics, it can be seen from Table 9 that with increasing GSF, the bitter, sour taste and specific aroma increased considerably, while the salty taste did not reach the limits of black bread PN = 1.58. The persistence of the flavor after chewing and swallowing increased considerably, from 1.66 (P0) to 4.37 (P4), exceeding the PN black sample values of 2.76.

In Figure 3, the sensory analysis of the studied bread samples is presented cumulatively. Each axis is representing a criterion of sensory analysis (sour taste, salty taste, etc.) using marks from 0 to 5. Joining the points corresponding to the marks given by the panelists for a certain test (Table 10) results in a specific contour with a certain color. The overlapping of the contours gives an overview of the enclosement of samples P1–P4, between the two control samples P0 and PN, with which the consumer is accustomed. It can be seen that for samples P1 (green color) and P2 (blue color), almost all the characteristics were between the control and limit samples, namely sample P0 (wheat flour bread, sample type 480, represented by the color red) and sample PN (sample of bread made of black wheat flour type 1250, represented in black).

Based on the submitted grades, the graph in Figure 4 was made, which shows that the most appreciated samples were those with 3% and 5% addition of GSF. Therefore, between standard samples P0 and PN, there were samples P1 and P2, the other samples having characteristics that do not recommend them for general use but possibly for medical or special purposes.

## 4. Discussion

The rheological analyses performed show a decrease in water absorption capacity with increasing GSF content, to values close to the lower limit of the optimal range of 55–62% for the manufacture of bakery products [64]. This shows that the maximum percentage of GSF addition (9%) was chosen correctly, similar values being obtained by Sporin et al. [48], Mildner-Szkudlarz et al. [65], and Kuchtova et al. [67]. Regarding the phenomenon of amylolysis of starch gel, the obtained values indicate an intensification of the enzymatic activity, as was observed also by Singh et al. [70] in a study of 15 samples.

Physicochemical indicators of breads with GSF addition meet the normal baking standards with samples P1 and P2, sample P3 having a porosity of 56.8%, lower than the acceptable minimum of 62% [71]. The P4 sample has a porosity of 50.8%, well below the mentioned limit.

From the sensorial point of view, samples P1 (3% GSF) and P2 (5% GSF) fall within the limits of the characteristics of the control samples P0 (wheat flour type 480, white bread) and PN (wheat flour type 1250, black bread) for 6 out of 10 criteria: specific aroma, salty taste, persistence of aroma after chewing and swallowing, core softness, crust color, and uniformity of the core pores. P3 sample (7% GSF) overlapped the limits of the control samples regarding the core color, core crumbliness, and sour and bitter taste.

The results of the nutritional analyses presented in Section 3.1 open up interesting opportunities to create bakery products that exceed a content of 3 g/100 g fiber and that can bear the nutritional mention of fiber source, rich in fiber according to European regulations for nutritional labeling. It is also possible to obtain bakery products with a mineral content higher than 15% of the RDI (with nutritional labeling “source of…”) and higher than 30% of the RDI (with nutritional labeling “rich in…”).

Table 11 summarizes for all samples studied the total fiber content (g/100 g) and mineral content (Ca, Mg, K, Cu, Fe, and Zn) expressed as a percentage of the reference nutritional value (800 mg for calcium; 375 mg for magnesium; 2000 mg for potassium; 1 mg for copper; 14 mg for iron, and 10 mg for zinc). The samples for which “source of…” type mentions can be made have been marked in yellow and the samples for which “rich in…” type mentions can be made have been marked in orange. Taking into account the results of the sensory analyses performed, the samples accepted by consumers and that can be widely used are marked in green. The samples that do not fully meet the bakery standards but could have medical nutritional purposes are marked in pink. The samples rejected by consumers but that have a special nutritional potential are marked in blue, the respective mixtures of flours being usable for the manufacture of products from the category of biscuits, sticks, pastas, etc. By adding 1–5% gluten in the manufacturing recipes, the volume and texture corresponding to the samples with special destination and medical use can be ensured, so these can be accepted by the large mass of consumers, thus becoming of commercial use.

According to the results obtained and presented in Table 9, the following can be concluded: From the four samples (P1–P4) with the addition of GSF, samples P1 and P2 have a content of more than 3 g of fiber and may be referred to as a “source of fiber”. They are also accepted from the sensorial point of view and can be classified as commercial products. Sample P3, although it meets the condition of 3 g/100 g and can be attributed the nutritional term “source of fiber,” from a sensorial point of view falls into the category of products with potential medical purpose.

The P4 sample, which has more than 6 g of fiber/100 g, can be given the nutritional mention “rich in fiber” but as it does not correspond from a sensory point of view, it is included in the category of special use products (food banks, military use, etc.). Regarding Mg content, only samples P3 and P4 can be assigned the nutritional term “magnesium source,” containing over 15% magnesium of the RDI, samples that from a sensory point of view fall into the category of products for medical use and special use. Regarding copper content, all four samples studied (P1–P4) have a content of more than 15% copper of the RDI, thus receiving the nutritional mention “copper source”. Regarding the zinc content, all samples with added GSF (P1–P4) contain more than 30% zinc of the RDI and thus can be termed “rich in zinc”.

## 5. Conclusions

Mixtures of wheat flour with added grape seed flour were noted for their high fiber content (between 4.47% and 9.27%) as well as a high mineral content for calcium (54.67 mg/100 g–76.40 mg/100 g) and magnesium (58.23 mg/100 g–79.24 mg/100 g). From rheological and sensorial points of view, the 7% and 9% replacement samples showed unsatisfactory behavior and are suitable as special products.

The results of the nutritional indicators of the bread samples showed that the studied flour mixtures provide a balanced intake of fiber, protein, and minerals, thus improving the nutritional quality of the main matrix (wheat flour). Increasing the degree of replacement of wheat flour with grape flour for the production of bakery products leads to declining rheological parameters and lower technological performance. The hydration capacity of the flour mixtures and the stability of the doughs were similar to those of the control sample. However, the volume of the baking samples and the mechanical and textural characteristics of the finished products were usually lower.

From a rheological point of view, the baking qualities deteriorated: the water absorbing capacity (CH) decreased from 58.2% to 55.8%, the dough stability increased from 8.50 min to 9.83 min, the α slope varied from −0.066 Nm/min to −0.104 Nm/min, the β slope increased from 0.576 Nm/min to 0.630 Nm/min, and the γ slope varied from −0.100 Nm/min to −0.198 Nm/min.

The results of the sensory analyses performed by the panel evaluators show that samples P1 (3% GSF) and P2 (5% GSF) were within the limits of the characteristics of the control samples P0 (wheat flour type 480, white bread) and PN (wheat flour type 1250, black bread) for: specific aroma, salty and bitter taste, persistence of aroma after chewing and swallowing, core softness, crust color, and uniformity of the core pores. The P3 sample (7% GSF) overlapped the limits of the control samples regarding the core color and the core crumbliness. Sour taste was significant for all samples with GSF addition, and bitter taste was significant for samples P3 (7% GSF) and P4 (9% GSF) due to the presence of tannins in grape seeds.

The decrease in the volume of samples can be explained by the phenomena of inhibition of amylase activity by phenolic compounds present in grape marc. As a result, the maltose content will be lower, with effects of a proportional decrease in the amount of carbon dioxide generated by the yeast during fermentation. Under these conditions, the bakery products obtained will have a low volume and a compact texture.

Increasing the GSF added led to an increase in the acidity of baking samples, with beneficial effects on their shelf life. Following the interpretation of the experimental results obtained, it can be concluded that all the baking samples with the addition of grape seed flour allow displaying nutritional information on the fiber, copper, and zinc content. The samples with a substitution degree of 7% and 9% of GSF allow, in addition, the nutritional mention “magnesium source.” Samples with 7% GSF do not meet the bakery standards but may have a medical purpose due to their special nutritional composition. The 9% baking samples can be used to design and make special-purpose products, such as sucking sticks and biscuits.

Experimental research on baking samples opens up interesting prospects for the use of results in medical gastronomy or personalized gastronomy, depending on the nutritional deficiencies of the concerned consumer, as well as in national and European food safety strategies.

## Figures and Tables

**Figure 1 foods-11-01589-f001:**
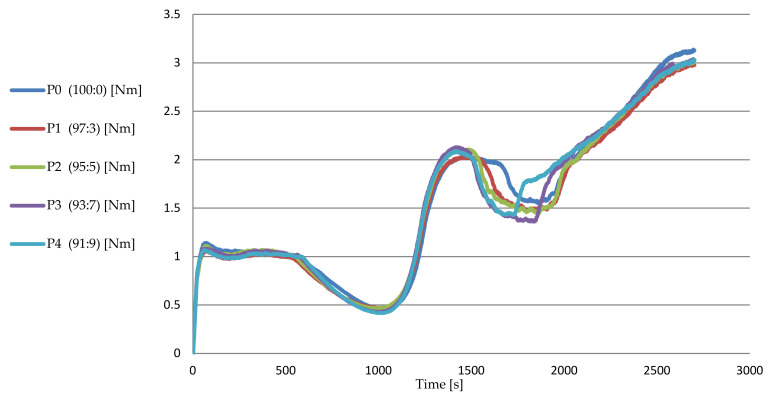
The influence of the substitution level of grape seed flour on the Mixolab curves.

**Figure 2 foods-11-01589-f002:**
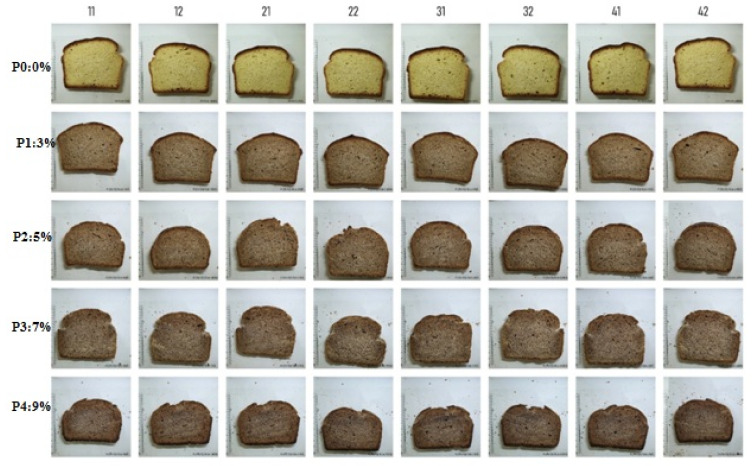
Bread slices of samples P0, P1, P2, P3, and P4.

**Figure 3 foods-11-01589-f003:**
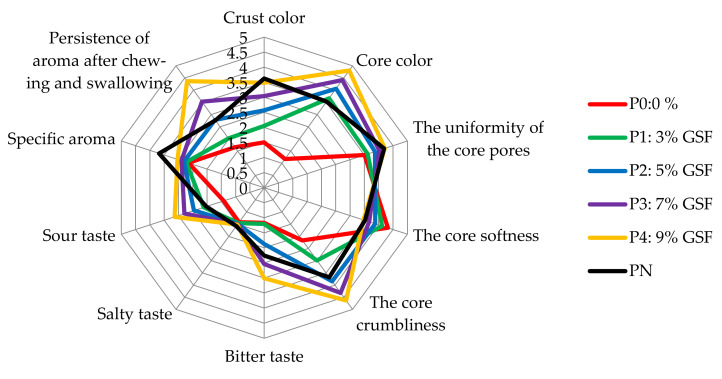
Results of sensory analysis of bread samples with the addition of GSF.

**Figure 4 foods-11-01589-f004:**
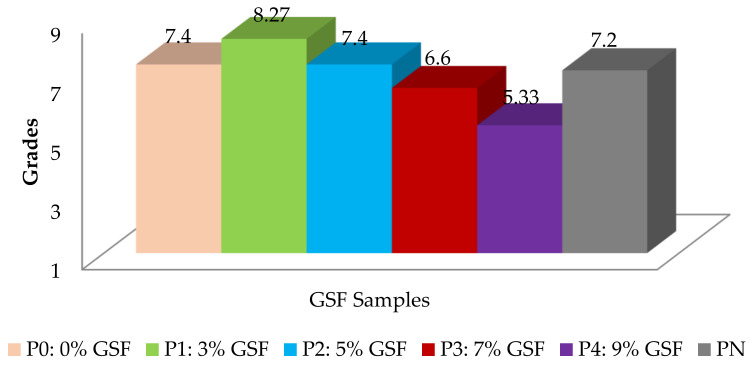
Results of the hedonic test of bread samples with the addition of GSF.

**Table 1 foods-11-01589-t001:** Coding of experimental samples with the addition of partially defatted grape seed flour.

Samples	Sample Composition (*w*/*w*)
P0	Sample 0–100% wheat flour type 480
P1	Sample 1–97% wheat flour type 480 + 3% defatted grape seed flour
P2	Sample 2–95% wheat flour type 480 + 5% defatted grape seed flour
P3	Sample 3–93% wheat flour type 480 + 7% defatted grape seed flour
P4	Sample 4–91% wheat flour type 480 + 9% defatted grape seed flour

**Table 2 foods-11-01589-t002:** Mixolab parameters correlation and significance.

Parameter	Calculation Method	Significance
Water Absorption (%)	Quantity of water required to obtain C1 = 1.1 Nm +/− 0.05	Amount of water taken up by flour to achieve the desired consistency and create a quality end-product.
Time for C1 (min)	Time required to obtain C1	Dough formation time: The stronger the flour, the longer it takes.
Stability (min)	Time during which torque is > C1–11% (constant T° phase)	Dough resistance to kneading: The longer it takes, the “stronger” the dough.
Amplitude (Nm)	Curve width at C1	Dough elasticity: The higher the value, the greater the dough elasticity.

Source: Mixolab Applications Handbook-May 2012 edition [55].

**Table 3 foods-11-01589-t003:** The technological process and the final recipe for obtaining the control sample P0.

Control Sample (P0)
Direct Method Recipe
Wheat flour type 480: 1.5 kg Yeast Pakmaya: 0.030 kg Salt: 0.015 kg Water: 1.050 L Kneading: 7 min. slowly and then 5 min. quickly in the mixer. Fermentation: 45 min in a fermenter (controlled atmosphere: 32 °C, relative humidity 70%). Portioning into 850 g pieces. Round format modeling; intermediate fermentation for 15 min; final form modeling. Final fermentation for 45 min (controlled atmosphere: 32 °C, relative humidity 70%). Baking in the oven for 35 min at 230 °C, with steam treatment in the first 10 s. Cooling to room temperature (minimum 2 h).

**Table 4 foods-11-01589-t004:** Chemical composition of the grape seed and wheat flours.

Parameter	Grape Seed Flour	CV (%)	Wheat Flour	CV (%)	*p*-Value (*t*-Test)
Moisture content (g/100 g)	9.2 ± 0.10 ^a^	0.714	12.6 ± 0.09 ^b^	1.087	<0.0001
Ash (g/100 g)	3.03 ± 0.13 ^a^	0.330	0.48 ± 0.01 ^b^	2.083	<0.0001
Protein (g/100 g)	16.32 ± 0.21 ^a^	0.429	12.01 ± 0.12 ^b^	0.999	<0.0001
Fat (g/100 g)	5.92 ± 0.19 ^a^	0.337	1.03 ± 0.05 ^b^	4.854	<0.0001
Raw fiber (g/100 g)	83.01 ± 0.78 ^a^	0.096	1.98 ± 0.12 ^b^	6.061	<0.0001
Sugars (g/100 g)	11.31 ± 0.11 ^a^	6.454	1.02 ± 0.13 ^b^	12.750	<0.0001
Potassium (mg/100 g d.m.)	360.23 ± 0.89 ^a^	0.247	187.13 ± 0.75 ^b^	0.401	<0.0001
Magnesium (mg/100 g d.m.)	397.81 ± 0.97 ^a^	0.243	47.73 ± 0.55 ^b^	1.152	<0.0001
Calcium (mg/100 g d.m.)	405.89 ± 1.17 ^a^	0.288	43.81 ± 0.59 ^b^	1.347	<0.0001
Iron (mg/100 g d.m.)	0.78 ± 0.02 ^a^	2.564	1.11 ± 0.02 ^b^	1.802	0.0132
Zinc (mg/100 g d.m.)	0.73 ± 0.07 ^a^	9.589	5.43 ± 0.22 ^b^	4.052	<0.0001
Copper (mg/100 g d.m.)	0.96 ± 0.1 ^a^	10.420	0.18 ± 0.02 ^b^	11.110	<0.0001

Note: Values are the means of triplicate determinations. The results are presented as mean values ± standard deviation. Different letters in the same row indicate significant differences (*p* < 0.05); CV, coefficient of variation.

**Table 5 foods-11-01589-t005:** The daily values of nutrients recommended per day (RDI) [60].

Constituents	RDI (FDA 2011) mg
Potassium	4700
Calcium	1000
Magnesium	400
Iron	18
Sodium	2400
Zinc	15
Manganese	4
Copper	2

Source: FDA. Available online: http://www.fda.gov/nutritioneducation (accessed on 23 March 2022).

**Table 6 foods-11-01589-t006:** Experimental results of the nutritional composition and mineral content of wheat flour mixtures with the addition of grape seed flour.

Parameter/Sample	P0 (0% GSF)	P1 (3% GSF)	P2 (5% GSF)	P3 (7% GSF)	P4 (9% GSF)
Ash (% d.m.)	0.48 ± 0.01	0.56 ± 0.01	0.61 ± 0.01	0.66 ± 0.02	0.71 ± 0.02
Protein (% d.m.)	12.01 ± 0.12	12.16 ± 0.1	12.22 ± 0.09	12.31 ± 0.01	12.47 ± 0.09
Fat (% d.m.)	1.03 ± 0.05	1.20 ± 0.06	1.31 ± 0.07	1.43 ± 0.06	1.55 ± 0.07
Raw fiber (% d.m.)	1.02 ± 0.13	4.47 ± 0.21	6.03 ± 0.32	7.65 ± 0.47	9.27 ± 0.56
Sugars (% d.m.)	1.98 ± 0.12	1.36 ± 0.13	1.53 ± 0.14	1.74 ± 0.15	1.95 ± 0.16
Potassium (mg/100 g d.m.)	187.13 ± 0.75	192.32 ± 0.88	195.79 ± 0.93	199.25 ± 1.02	202.71 ± 1.11
Magnesium (mg/100 g d.m.)	47.73 ± 0.55	58.23 ± 0.68	65.23 ± 0.71	72.24 ± 0.78	79.24 ± 0.79
Calcium (mg/100 g d.m.)	43.81 ± 0.59	54.67 ± 0.89	61.91 ± 0.96	69.16 ± 1.07	76.40 ± 1.29
Iron (mg/100 g d.m.)	1.11 ± 0.02	1.10 ± 0.02	1.09 ± 0.02	1.08 ± 0.02	1.07 ± 0.02
Zinc (mg/100 g d.m.)	5.43 ± 0.22	5.29 ± 0.20	5.20 ± 0.20	5.10 ± 0.18	5.01 ± 0.16
Copper (mg/100 g d.m.)	0.18 ± 0.09	0.20 ± 0.10	0.22 ± 0.11	0.23 ± 0.12	0.25 ± 0.13

Note: The values obtained are the average of three consecutive determinations ± SD.

**Table 7 foods-11-01589-t007:** Rheological characteristics of 100% wheat flour and flour mixtures.

	P0 (100% Wheat Flour)	P1 (97% Wheat Flour + 3% Grape Seed Flour)	P2 (95% Wheat Flour + 5% Grape Seed Flour)	P3 (93% Wheat Flour + 7% Grape Seed Flour)	P4 (91% Wheat Flour + 9% Grape Seed Flour)
Water Absorption (%)	58.2 ± 0.06 ^a^	58.0 ± 0.06 ^a^	56.7 ± 0.1 ^b^	56.2 ± 0.11 ^c^	55.8 ± 0.08 ^d^
Stability (min)	8.50 ± 0.33 ^a^	9.10 ± 0.26 ^a,b^	9.26 ± 0.18 ^b,c^	9.48 ± 0.13 ^b,c^	9.83 ± 0.22 ^b,c,d^
Amplitude (Nm)	0.100 ± 0.01 ^a^	0.104 ± 0.01 ^a^	0.081 ± 0.01 ^a^	0.078 ± 0.01 ^a,b^	0.075 ± 0.01 ^a,b^
α	−0.066 ± 0.003 ^a^	−0.074 ± 0.002 ^b^	−0.086 ± 0.002 ^c^	−0.094 ± 0.003 ^d^	−0.104 ± 0.002 ^e^
β	0.576 ± 0.003 ^a^	0.588 ± 0.004 ^b^	0.604 ± 0.005 ^c^	0.618 ± 0.004 ^d^	0.630 ± 0.005 ^e^
γ	−0.100 ± 0.002 ^a^	−0.118 ± 0.003 ^b^	−0.136 ± 0.003 ^c^	−0.164 ± 0.010 ^d^	−0.198 ± 0.010 ^e^
C1	1.143 ± 0.01 ^a^	1.065 ± 0.03 ^b^	1.074 ± 0.01 ^b^	1.081 ± 0.02 ^b^	1.068 ± 0.03 ^b^
C2	0.461 ± 0.02 ^a^	0.469 ± 0.01 ^a^	0.468 ± 0.01 ^a^	0.428 ± 0.02 ^a^	0.416 ± 0.03 ^a^
C3	2.041 ± 0.01 ^a^	2.090 ± 0.02 ^a,b^	2.106 ± 0.01 ^a,b^	2.129 ± 0.01 ^a,b^	2.211 ± 0.02 ^b^
C4	1.553 ± 0.01 ^a^	1.479 ± 0.01 ^b^	1.458 ± 0.01 ^b,c^	1.436 ± 0.02 ^c^	1.429 ± 0.02 ^c^
C5	3.131 ± 0.09 ^a^	3.083 ± 0.1 ^a^	3.009 ± 0.12 ^a^	2.873 ± 0.08 ^a,b^	2.632 ± 0.11 ^b^

Note: Values are the means of triplicate determinations. The results are presented as mean values ± standard deviation. Different letters in the same row indicate significant differences (*p* < 0.05).

**Table 8 foods-11-01589-t008:** Experimental results obtained from rheological and enzymatic analyses performed with the Mixolab apparatus Chopin + protocol for the five flour samples.

Mixolab Parameters, Chopin + Protocol		Samples of Flour Mixtures				
		P0	P1	P2	P3	P4
Phase 1	C1 (Nm)	1.143 ± 0.01 ^a^	1.065 ± 0.03 ^b^	1.074 ± 0.01 ^b^	1.081 ± 0.02 ^b^	1.068 ± 0.03 ^b^
	TC1 (min)	1.20 ± 0.1 ^a^	1.22 ± 0.08 ^a^	1.12 ± 0.07 ^b^	1.09 ± 0.1 ^b^	1.05 ± 0.1 ^c^
Phase 2	C2 (Nm)	0.461 ± 0.02 ^a^	0.469 ± 0.01 ^a^	0.468 ± 0.01 ^a^	0.428 ± 0.02 ^a^	0.416 ± 0.03 ^a^
	TC2 (min)	16.62 ± 0.13 ^a^	16.68 ± 0.11 ^a^	16.40 ± 0.08 ^a^	16.63 ± 0.12 ^a^	17.00 ± 0.08 ^a^
C1-C2 (Nm)		0.682 ± 0.01 ^a^	0.596 ± 0.02 ^b^	0.606 ± 0.01 ^b^	0.653 ± 0.01 ^a^	0.652 ± 0.02 ^a^
Phase 3	C3 (Nm)	2.041 ± 0.01 ^a^	2.090 ± 0.02 ^a,b^	2.106 ± 0.01 ^a,b^	2.129 ± 0.01 ^a,b^	2.211 ± 0.02 ^b^
	TC3 (min)	24.75 ± 0.48 ^a^	24.30 ± 0.30 ^a^	24.08 ± 0.24 ^a^	23.63 ± 0.05 ^a^	23.62 ± 0.04 ^a^
C3-C2 (Nm)		1.58 ± 0.01 ^a^	1.621 ± 0.01 ^b^	1.638 ± 0.01 ^b^	1.701 ± 0.02 ^b^	1.795 ± 0.02 ^c^
Phase 4	C4 (Nm)	1.553 ± 0.01 ^a^	1.479 ± 0.01 ^b^	1.458 ± 0.01 ^b,c^	1.436 ± 0.02 ^c^	1.429 ± 0.02 ^c^
	TC4 (min)	30.95 ± 0.18 ^a^	30.67 ± 0.12 ^a^	29.95 ± 0.10 ^a^	29.43 ± 0.08 ^a,b^	28.75 ± 0.06 ^b^
C3-C4 (Nm)		0.488 ± 0.01 ^a^	0.611 ± 0.01 ^b^	0.648 ± 0.01 ^b^	0.693 ± 0.02 ^c^	0.782 ± 0.02 ^d^
Phase 5	C5 (Nm)	3.131 ± 0.09 ^a^	3.083 ± 0.10 ^a^	3.009 ± 0.12 ^a^	2.873 ± 0.08 ^a,b^	2.632 ± 0.11 ^b^
	TC5 (min)	45.02 ± 0.01 ^a^	45.02 ± 0.01 ^a^	45.02 ± 0.01 ^a^	45.02 ± 0.01 ^a^	45.02 ± 0.01 ^a^
C5-C4 (Nm)		1.578 ± 0.01 ^a^	1.604 ± 0.01 ^a^	1.551 ± 0.01 ^b^	1.437 ± 0.01 ^c^	1.203 ± 0.01 ^d^

Note: Values are the means of triplicate determinations. The results are presented as mean values ± standard deviation. Different letters in the same row indicate significant differences (*p* < 0.05).

**Table 9 foods-11-01589-t009:** Physicochemical indicators of experimental breads with the addition of GSF.

Sample	Mass (kg)	Specific Volume (cm^3^/100 g)	Porosity (%)	Elasticity (%)	Flavor	Humidity (%)	Acidity (degree)
P0: 0%	0.680	290	84.3	95	Pleasant, characteristic of well-baked white bread	43.7	1.2
P1: 3%	0.675	278	82	92	Pleasant, similar to black bread, well baked	42.3	1.6
P2: 5%	0.681	220	61	92	Aroma of wine, sticky to chew	41.1	1.9
P3: 7%	0.672	208	56.8	87	Strong smell and taste of wine, sticky to chew, a slight feeling of sand, annoying	39.8	2.0
P4: 9%	0.674	197	50.8	83	Strong odor and taste of yeast, sticky to chew, intense sandy feeling, annoying	37.6	2.2

**Table 10 foods-11-01589-t010:** Summary of scores obtained by the panel evaluation of bread samples with the addition of GSF.

Sensorial Attribute	Crust Color	Core Color	The Uniformity of the Core Pores	The Softness of the Core	The Core Crumbliness	Bitter Taste	Salty Taste	Sour Taste	Specific Aroma	Persistence of Aroma after Chewing and Swallowing
Samples										
P0: 0%	1.51	1.17	3.51	4.32	2.16	1.17	1.42	1.42	2.66	1.66
P1: 3%	2.05	3.67	3.62	4.12	2.99	1.21	1.46	2.12	2.73	2.02
P2: 5%	2.57	4.06	3.87	3.88	3.84	1.87	1.49	2.46	2.81	2.78
P3: 7%	3.04	4.42	4.02	3.67	4.32	2.54	1.50	2.79	2.9	3.53
P4: 9%	3.48	4.81	4.21	3.47	4.63	3.01	1.52	3.14	3.03	4.37
PN	3.63	3.53	4.19	3.52	3.68	2.26	1.58	2.03	3.68	2.76

**Table 11 foods-11-01589-t011:** Synthesis of nutritional indicators for the studied baking samples.

Quality Indicator		Total Fiber (g/100 g)		Ca (% of RDI)	Mg (% of RDI)		K (% of RDI)	Cu (% of RDI)		Fe (% of RDI)	Zn (% of RDI)	
P0: control sample WF type 480		1.46		3.95	10.54		7.29	14.00		5.86	“rich in Zn”	38.8 **
P1: 97% WF + 3% GSF ^a^	Commercial use	“source of fiber”	3.51 *	4.95	12.74		7.52	“source of Cu”	15.00 *	6.00		37.1 **
P2: 95% WF + 5% GSF ^a^			4.24 *	5.51	12.94		7.78		16.00 *	6.07		37.4 **
P3: 93% WF + 7% GSF ^b^	Medical use		5.54 *	6.68	“source of Mg”	15.01 *	8.42		17.00 *	6.21		37.7 **
P4: 91% WF + 9% GSF ^c^	Special use	“rich in fiber”	7.37 **	7.43		15.83 *	8.59		19.00 *	6.43		39.8 **

Note: * 15% of the RDI, “source of…”; ** 30% of the RDI, “rich in…”; ^a^—commercial use; ^b^—medical use; ^c^—special use (food banks, military use, etc.).

## Data Availability

The data used in this study are available in this article.

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
