# Peer review of "Research on the Potential Use of Grape Seed Flour in the Bakery Industry"

_foods, 2022, doi:10.3390/foods11111589_

Round 1

Reviewer 1 Report

This work cover the use of grapes seed flour in preparation of bakery products. The results are interesting and could be used in development of nutritive and novel food items. This manuscript need changes which are mention below

Abstract: The abstract is not informative enough. Please provide numerical values of the results with significant difference. There are grammatical errors and need to minimize the sentences in length.

Keywords: Good enough

Introduction: The similar font size should maintain throughout the manuscript. Check page 2 lines 81-84

How to control the composition of wheat flour?

Please provide pervious study upon grapes seed utilization.

What is the average production of such by-products? Page 1 lines 40-42. Mention in tons here

Please rewrite the lines from 81-93. The format and pervious study is not enough, secondly the references are not in Journal format

Methods and materials: What is degreased grapes seed?

What was the mixing ration? Is wheat flour and grapes seed were mixed w/w or w/v? Please explain in methods.

What is composition of wheat flour? The resultant flour from mixing with grapes seed seems to develop confusion among the final compositions values. It seem the recorded values showed in this manuscript are only for wheat flour.

What type of rheological parameters were considered in rheological studies?

Is any equation was used for rheological parameters?

Crude fiber composed of soluble and insoluble dietary fiber? Please explain

Results and discussion: What is figure 3? Normal reader may difficult to understand such figure, please change the presentation of this figure or provide table if possible.

How the panel evaluation was conducted for sensory evaluation?

Over all the discussion part is not enough for these results, I suggest to improve the discussion part in depth with update references.

Conclusion: please change conclusion part after modification in discussion.

Author Response

Point 1. Abstract: The abstract is not informative enough. Please provide numerical values of the results with significant difference. There are grammatical errors and need to minimize the sentences in length.

Response 1: The abstract was changed as following:

Grape seeds are one of the most accessible by-products of the wine industry, in large quantities (about 2.4 million t / year). Numerous researchers have shown that grape seeds have a high potential for use as a functional ingredient in the food industry due to their high content of protein, fiber, minerals and polyphenols. The aim of the paper is to evaluate the possibilities of using grape seed flour (GSF) in bakery industry, from both chemical and rheological point of view. Research shows that grape seed flour contains about 42 times more fiber than wheat flour, approximately 9 times more calcium, 8 times more magnesium and 2 times more potassium. To assess this potential, four samples of bread from flour mixtures with 3%, 5%, 7% and 9% (w / w) degree of replacement with GSF were performed and analyzed, which were compared with a control sample from 100% wheat flour. From a rheological point of view, the baking qualities are deteriorating: water absorber capacity (CH) decreases from 58.2% to 55.8%, the dough stability increases from 8.50 min to 9.83 min, α slope varies from -0.066 Nm / min to -0.104 Nm / min, β slope increases from 0.576 Nm / min to 0.630 Nm / min, and É£ slope varies from -0.100  Nm / min to -0.198 Nm / min. The sensory analyzes performed by the panel of evaluators enclosed the samples with 3% and 5% GSF between the two control samples from flour type 480 and 1250. The conclusions show that the sample containing 7% and 9% are unsatisfactory from a rheological and sensorial point of view, and the samples with 3% and 5% can receive mentions of fibre source and Cu source, respectively rich in Zn.

Point 2. Keywords: Good enough

Response 2: The keywords „bread” andnutritional properties” were also added.

Point 3. Introduction: The similar font size should maintain throughout the manuscript. Check page 2 lines 81-84

Response 3: The font was changed to the one used in the Foods Journal template

Point 4. How to control the composition of wheat flour?

Response 4: The characteristics of wheat flour for baking are specified in the SR 878/1996,  the flour used in our experiments respecting this standard.

Point 5. Please provide pervious study upon grapes seed utilization.

Response 5: The following previous studies were added:

Iriondo-DeHond et al. [42] have studied the use of winery byproducts extracts (grape pomace, seed and skin) and a mixture of inulin-type fructans (inulin and FOS) as new ingredients for the development of yogurts with antioxidant and antidiabetic properties.

Libera et al. [43] have used grape seed extract in the meat industry as a natural antioxidant instead of sodium ascorbate, with very optimistic results.

Amoah et al. [44] have presented the possibilities of functional bread development through it’s valorization with certain plant based by-products among which grape pomace and grape seed extracts. An increased bioactivity of functional bread has been observed by incorporation of flour from the plant-based by-products. In most cases, bread enriched with up to 6% flour from by-products had enhanced qualities. Regarding sensory acceptability of bread, formulations up to an average of 5% resulted in bread with acceptable organoleptic perceptions from consumers.

Tremlova et al. [45] have obtained “vegan” sausages with the addition of grape seed flour (GSF)  in different concetrations (0%, 1%, 3%, 7%, 10%, and 20%) The results indicated that the GSF addition resulted in a higher antioxidant capacity of experimentally produced vegan sausages. Regarding the sensory evaluation, vegan sausages with 1% (according to taste evaluation) and 3% additions of GSF were the most acceptable by panelists.  Also the sustainability of GSF usage is achieved, since it is a waste material generated worldwide within winemaking technology.

Král et al. [46] have studied the addition of selected herbs and spices (ground cloves, cinnamon, mint, and grape flour) in biscuits and the content of polyphenols and antioxidant capacity was measured, as well as their sensory properties and attractiveness to consumers. The results showed an increased antioxidant capacity for all samples, as well as for polyphenols. At a 3% addition of the selected herbs and spices have confirmed an overall acceptability from consumers. The conclusion based on the measurements is that a reasonable addition of natural substances containing natural antioxidants improve the overall quality of final products, in this case, biscuits.

Point 6. What is the average production of such by-products? Page 1 lines 40-42. Mention in tons here

Response 6: According to a study conducted by Heuze and Tran in 2017 [3], within the program „Feedipedia” that is part of INRAE, CIRAD, AFZ and FAO, seeds account for up to 6% of berry weight. The most important producers in the world are China (9.6 million t), the USA (6.6 million t), Italy (5.8 million t), France (5.3 million t), Spain (5.2 million t), Turkey (4.2 million t), Chile (3.2 million t), Argentina (2.8 million t) and Iran (2.1 million t). The EU accounts for 75% of grape production and 57% of wine production. The main wine producers are Italy, France, Spain (where most of the grape production is used for wine) and the USA.  Assuming a worldwide production of 40 million t of fresh grapes used for wine, with a seed proportion of 1-6%, the potential amount of grape seeds can be estimated between 0.4 and 2.4 million t.

Point 7. Please rewrite the lines from 81-93. The format and pervious study is not enough, secondly the references are not in Journal format

Response 7: The format of the mentioned text, and the references were changed according to the template format throughout the manuscript. Previous studies that were mentioned at point 5 were added.

Point 8. Methods and materials: What is degreased grapes seed?

Response 8: The word degreased was changed throughout the manuscript with the word defatted.

Point 9. What was the mixing ration? Is wheat flour and grapes seed were mixed w/w or w/v? Please explain in methods.

Response 9: The ratio of the studied flours were w/w. The w/w mixing ration was added into the manuscript (abstract and table 1 where the mixtures are presented).

Point 10. What is composition of wheat flour? The resultant flour from mixing with grapes seed seems to develop confusion among the final compositions values. It seem the recorded values showed in this manuscript are only for wheat flour.

Response 10: The composition of wheat flour was presented in table 4, but for a better understanding the following table has been introduced:

Table 6. Experimental results on the nutritional composition and mineral content of wheat flour mixtures with the addition of grape seed flour

Parameter / Sample

P0 (0% GSF)

P1 (3% GSF)

P2 (5% GSF)

P3 (7% GSF)

P4 (9 % GSF)

Ash (% d.m)

0.48 ± 0.01

0.56 ± 0.01

0.61± 0.01

0.66 ± 0.02

0.71± 0.02

Protein (% d.m)

12.01± 0.12

12.16 ± 0.1

12.22± 0.09

12.31± 0.01

12.47± 0.09

Fat (% d.m)

1.03 ± 0.05

1.20 ± 0.06

1.31± 0.07

1.43 ± 0.06

1.55 ± 0.07

Raw fiber (% d.m)

1.02 ± 0.13

4.47 ± 0.21

6.03 ± 0.32

7.65 ± 0.47

9.27 ± 0.56

Sugars (% d.m)

1.98 ± 0.12

1.36 ± 0.13

1.53 ± 0.14

1.74 ± 0.15

1.95 ± 0.16

Potassium ( mg/100g d.m)

187.13 ± 0.75

192.32 ± 0.88

195.79 ± 0.93

199.25 ± 1.02

202.71 ± 1.11

Magnesium ( mg/100g d.m)

47.73 ± 0.55

58.23 ± 0.68

65.23 ± 0.71

72.24 ± 0.78

79.24 ± 0.79

Calcium ( mg/100g d.m)

43.81± 0.59

54.67 ± 0.89

61.91 ± 0.96

69.16 ± 1.07

76.40 ± 1.29

Iron ( mg/100g d.m)

1.11 ± 0.02

1.10 ± 0.02

1.09 ± 0.02

1.08 ± 0.02

1.07 ± 0.02

Zinc ( mg/100g d.m)

5.43 ± 0.22

5.29 ± 0.20

5.20 ± 0.20

5.10 ± 0.18

5.01 ± 0.16

Copper ( mg/100g d.m)

0.18 ± 0.09

0.20 ± 0.10

0.22 ± 0.11

0.23 ± 0.12

0.25 ± 0.13

Note: The values ​​obtained are the average of 3 consecutive determinations; ± SD

In table 6 are presented the experimental results on the nutritional composition and mineral content of wheat flour mixtures with the addition of GSF. The addition of GSF resulted in significant increases in all 4 samples of mixtures in terms of fiber content, as follows: for sample P1 a relative increase of 125.75% was identified, for sample P2 the relative increase was 204.54%, sample P3 registered a relative increase of 286.36% and in the case of the last samples, P4 the relative increase was 368.18%. At the same time, there were significant increases in sugar content, 91.17% for the P4 sample (addition of 9% GSF), an increase in fat content of more than 50% and a slight increase (3.83%) of the protein content, proportional with the increase of the GSF addition.

Regarding the content of mineral substances for each of the studied samples P0 ... P4 the following are presented.

Significant increases are recorded in the Ca content, where an increase can be observed from the value of 43.81 mg / 100 g (sample P0) to the value of 76.40 mg / 100 g (sample P4), respectively an increase of 74.38%. Also, there are increases of over 60% in the case of Mg content, respectively 38.88% for Cu content. Increases below 10% are recorded for the K content. As for the Fe content, it decreased by 3.6% (P0: 1.11 mg Fe / 100 g, P4: 1.07 mg Fe / 100 g), and Zn content decreased by 7.73% (P0: 5.43 mg Zn / 100 g, P4: 5.01 mg Zn / 100 g).

Point 11. What type of rheological parameters were considered in rheological studies?

Response 11: The studied rheological parameters refer to the dynamic viscosity of the sample, evaluated by the torsion moment, measured by the kneading arms of a horizontal micro mixer, which is part of the Mixolab equipment. During the kneading of the flour and water mixture, the hydration of the proteins takes place and the viscosity of the dough increases due to the formation of the protein chains, respectively of the gluten. Next, the mixer tank is heated, the protein chains are denatured under the influence of heat and the mechanical effect of the kneading arms. The diagram obtained by measuring the torque moment of the kneading arms during hydration, heating and cooling of the dough, has five specific points (C1 ... C5) that allow the evaluation of parameters such as the hydration capacity of flour, gluten quality, amylolytic enzyme activity. The Mixolab device together with the „Chopin +” protocol has become an international standard (ICC 173), incorporating in a single analysis a cumulation of evaluations made with the classic equipments such as farinograph, amylograph and alveograph.

Point 12. Is any equation was used for rheological parameters?

Response 12: No, the main parameters are generated by the Mixolab equipment, based on the torque diagram of the kneading arms during hydration, heating and cooling of the dough.

Point 13. Crude fiber composed of soluble and insoluble dietary fiber? Please explain

Response 13: Crude fiber is the insoluble part of the cell wall. It is obtained as a insoluble residue of an acid hydrolysis, followed by an alkaline one.

Point 14. Results and discussion: What is figure 3? Normal reader may difficult to understand such figure, please change the presentation of this figure or provide table if possible.

Response 14: In figure 3 on each axis were represented the criteria of sensory analysis (sour taste, salty taste, etc.) by marks from 0 to 5. Joining the points corresponding to the marks given by the panelists for a certain test, result in the specific contour with a certain color. The overlapping of the contours gives us an overview of the framing of samples P1, P2, P3, P4, between the 2 control samples P0 and PN, with which the consumer is accustomed. In figure 3 are presented the main sensorial criterias with marks from 0 to 5 (representing the average of the marks given by the panelists), as it is presented also in table 10.

Point 15. How the panel evaluation was conducted for sensory evaluation?

Response 15: The sensory evaluation was conducted as it is presented in chapter 2.8.

A group of 10 specially trained panelist, with ages between 25 and 60, evaluated the bread samples, giving grades from 1 (lowest intensity) to 5 (highest intensity), for the following sensory attributions: crust color (degree of perceived brown color characterizing the crust), crumb color (degree of color darkness in the crumb ranging from white to dark brown), crumb pore uniformity (size of pores on the surface; (small/big), crumb softness (minimum force necessary to compress the sample), bitter taste (perceived by the back of the tongue and characterized by solutions of quinine, caffeine, and other alkaloids; usually caused by over-roasting), salty taste (fundamental taste sensation elicited by sodium chloride), sour taste (fundamental taste sensation evoked by acids, e.g., tartaric acid), specific aroma (aroma of fresh baked bread and odor associated with aromatic exchange from yeast fermentation), after-taste (flavor staying after tasting). Also, there has been made a consumer overall acceptability determination in a 9-point hedonic scale (from 9 = i like it extremely to 1 = i dislike it extremely), where 35 untrained panelists with ages between 21 and 60 (70% females and 30% males) have tasted the samples that were coded with 3 random letters in order to not influence their perception, and the results were expressed as mean. The main conditions in choosing the panelists were for them to not be smokers and to have a good health condition. All bread samples of flour mixtures P1-P4 were compared to the standards of 2 control samples P0 (wheat flour type 480 – white bread) and PN (wheat flour type 1250 – black bread).

Point 16. Over all the discussion part is not enough for these results, I suggest to improve the discussion part in depth with update references.

Response 16: The following were added:

The rheological analyzes performed show a decrease in water absorption capacity with increasing GSF content, to values ​​close to the lower limit of the optimal range of 55-62% for the manufacture of bakery products. [64]. This shows that the maximum percentage of 9% GSF addition was chosen correctly, similar values ​​being obtained by Sporin et al. [48], Mildner-Szkudlarz et al. [65] and Kuchtova et al. [67]. Regarding the phenomena of amylolysis of starch gel, the obtained values indicate a intensification of the enzymatic activity, as it was observed by Singh et al. [70] in a study of 15 samples.

Physico-chemical indicators of breads with GSF addition meet the normal baking standards with samples P1 and P2, sample P3 having a porosity of 56.8%, lower than the acceptable minimum of 62% [71]. The P4 sample has a porosity of 50.8%, well below the mentioned limit.

From the sensoryal point of view, samples P1 (3% GSF) and P2 (5% GSF) fall within the limits of the characteristics of the control samples P0 (wheat flour type 480 – white bread) and PN (wheat flour type 1250 – black bread) for 6 out of 10: specific aroma, salty taste, persitance of aroma after chewing and swallowing, core softness, crust color and uniformity of the core pores. P3 sample (7% GSF)  overlaped the limits of the control samples regarding the core color, core crumbliness, sour and bitter taste.

The results of the nutritional analyzes presented in Chapter 3.1 open up interesting opportunities to create bakery products that exceed a content of 3 g / 100 g fiber and that can bear the nutritional mention of fiber source, respectively rich in fiber according to European regulations for nutritional labeling. It is also possible to obtain bakery products with a mineral content higher than 15% of RDI (with nutritional labeling "source of ...") and respectively higher than 30% of RDI (with nutritional labeling "rich in ... ”).

Point 17. Conclusion: please change conclusion part after modification in discussion.

Response 17:  The following were added:

From a rheological point of view, the baking qualities are deteriorating: water absorber capacity (CH) decreases from 58.2% to 55.8%, the dough stability increases from 8.50 min to 9.83 min, α slope varies from -0.066 Nm / min to -0.104 Nm / min, β slope increases from 0.576 Nm / min to 0.630 Nm / min, and É£ slope varies from -0.100  Nm / min to -0.198 Nm / min.

The results of the sensory analyzes performed with the panel evaluators show that samples P1 (3% GSF) and P2 (5% GSF) fall within the limits of the characteristics of the control samples P0 (wheat flour type 480 – white bread) and PN (wheat flour type 1250 – black bread) for: specific aroma, salty and bitter taste, persistance of aroma after chewing and swallowing, core softness, crust color and uniformity of the core pores. P3 sample (7% GSF) overlaped the limits of the control samples regarding the core color and core crumbliness. Sour taste was significant for all samples with GSF addition, and bitter taste was significant for samples P3 (7% GSF) and P4 (9% GSF), due to the presence of tanins in grape seeds.

Reviewer 2 Report

Manuscript Number: foods-1739811, titled:

 Research on the potential use of grape seed flour in the bakery industry

Review 1 – 18 May 2022

Dear Editor of Foods

the argument is interesting but the manuscript needs to be improved. The introduction section has to be improved and well argued. The bibliography has to be improved. The References section contains inaccuracies and has to be completed and arranged using the instructions for authors of Foods, I have listed some correction to do in this section but the Authors have to carefully verify line by line.

I suggest a major revision

To the Authors (in detail):

  • the argument is interesting but the manuscript needs to be improved. The introduction section has to be improved and well argued. The bibliography has to be improved. The References section contains inaccuracies and has to be completed and arranged using the instructions for authors of Foods, I have listed some correction to do in this section but the Authors have to carefully verify line by line;

  • Introduction section, line 44 and in the whole manuscript, please include the bibliography as suggested by the instructions for authors of Foods, see the template and use also some recently published paper as a template;

  • Introduction section, lines 47-48, please, extend this sentence and discuss molecules in seeds also in relation of grape varieties;

  • Introduction section, lines 91-93 and in the whole manuscript, please, separate the numeric value from the unit: 3 g and not 11.3g;

  • Introduction section, the aim of this work is missed;

  • Introduction section. This section has to be improved, you have to explain about the interest for functionalized bakery products and support this statement with some recent published paper. In addition, you have proposed your manuscript to Foods, but only one reference you have included from Foods. Please, find, read and discuss: [X1, X2, X3].

[X1] Formulation of biscuits fortified with a flour obtained from bergamot by-products (Citrus bergamia, Risso).

Foods 2022, 11 (8), 1137.

https://doi.org/10.3390/foods11081137

[X2] Effects of shortening replacement with extra virgin olive oil on the physical–chemical–sensory properties of Italian Cantuccini biscuits.

Foods 2022, 11, 299

https://doi.org/10.3390/foods11030299

[X3] Enhancement of functional and nutritional properties of bread using a mix of natural ingredients from novel varieties of flaxseed and lupine.

LWT 91, 48-54 (2018).

  • Introduction section line 81 and in the whole manuscript: Mironeasa et al… have investigated. It is a group of researchers;
  • Caption of table 1: Table and not table;
  • Table 1 in the caption you have written defatted, in the table you have written degreased. Be consistent in the whole manuscript, I suggest defatted;
  • 2 section and in the whole manuscript, tables and figures, when you indicate the temperature, separate the numeric value from the symbol: 40-65 °C and not 40-65°C;
  • 3 sub-section, line 118 and in the whole manuscript: separate decimal by a dot and not by a comma;
  • 5 sub-section, line 149: what 80 is? Rpm or what? In addition, write -1 as an exponent (if you are meaning … per minute);
  • 5 sub-section, line 173 and in the whole manuscript, please, use the correct type for beta (β) from the ancient Greek alphabet, the type you have used is from the modern German alphabet and it is not beta;
  • Table 3, separate decimal by a dot and not by a comma;
  • 8 sub-section and in the whole manuscript. When you have indicated the significance you have used different criterion for spacing: p < 0.05 or p<0.05?, In addition p italicized or not?
  • Line 263 and in the whole manuscript, do not insert a dot after g. The dot after g only at the end of a sentence
  • Caption of table 5, insert one space after .. day;
  • Discussion section (Magnesium), Conclusions section (magnesium). Please be consistent in the whole manuscript for this and for all chemical compounds;
  • References section. This section has to be arranged in light of the instructions for authors of Foods. For example, the journal name has to be abbreviated (ref 15). The capital letter only for the first type. Please verify in the template;
  • The references section is not arranged as requited by Foods;
  • References section, ref 28: ect?

  • Please, very important, write in blue color or evidence differently the corrections you will do.

I suggest a major revision

Regards.

Author Response

Point 1. the argument is interesting but the manuscript needs to be improved. The introduction section has to be improved and well argued. The bibliography has to be improved. The References section contains inaccuracies and has to be completed and arranged using the instructions for authors of Foods, I have listed some correction to do in this section but the Authors have to carefully verify line by line;

Response 1: All highlighted issues regarding introduction section and bibliography improvement were done. Inaccuracies in the reference list have been fixed and several new bibliographical titles were added.

Point 2. Introduction section, line 44 and in the whole manuscript, please include the bibliography as suggested by the instructions for authors of Foods, see the template and use also some recently published paper as a template;

Response 2: The changes were made as suggested;

Point 3. Introduction section, lines 47-48, please, extend this sentence and discuss molecules in seeds also in relation of grape varieties;

Response 3: The following text was introduced:

Tita et al. have studied seven grape varieties and concluded that the phenolic compounds vary depending on the nature of the compound and the variety from which the seeds belong to. For example the richest varieties in syringic acid are Syrah and Novac (between 121.22 and 136.66 mg/L), followed by Burgund Mare and Cadarca. The lowest values are observed in the case of Cabernet Sauvignon and Merlot extracts. Gallic acid and vanillic acid are found most significantly  in the Novac variety—39.22 and 20.91 mg/L. Epicatechin gallate varies from 1.84 to 2.56 mg/L, the most significant values being in the case of seed extract from the Pinot noir variety. Caffeic acid is reaching a maximum of 1.56 mg/L, specific to the Novac variety. Resveratrol is found in the seeds of all varieties with values between 1.91 and 2.92 mg/L for Cabernet Sauvignon, 1.41 and 2.23 mg/L for Merlot, 1.93 and 2.37 mg/L for Pinot noir, 1.66 and 1.88 mg/L for Burgund Mare, 1.71 and 2.46 mg/L for Cadarca, 2.12 and 2.34 mg/L for Syrah and 2.16 and 2.38 mg/L for Novac variety.

Point 4. Introduction section, lines 91-93 and in the whole manuscript, please, separate the numeric value from the unit: 3 g and not 11.3g;

Response 4: All numerical values were separated from the units throughout the manuscript;

Point 5. Introduction section, the aim of this work is missed;

Response 5: The following sentence was added: The aim of this work is to evaluate the possibilities of using grape seed flour (GSF) in the bakery industry, as a functional ingredient, from both chemical and rheological point of view. Also, the maximum replacement degree  of wheat flour with GSF will be determined in the conditions of fulfilling the bakery standards and taking into account the sensory analyzes of the panelists. Finally, nutrition labeling recommendations will be made based on the results of nutritional analyzes.

Point 6. Introduction section. This section has to be improved, you have to explain about the interest for functionalized bakery products and support this statement with some recent published paper. In addition, you have proposed your manuscript to Foods, but only one reference you have included from Foods. Please, find, read and discuss: [X1, X2, X3].

[X1] Formulation of biscuits fortified with a flour obtained from bergamot by-products (Citrus bergamia, Risso). Foods 2022, 11 (8), 1137. https://doi.org/10.3390/foods11081137

[X2] Effects of shortening replacement with extra virgin olive oil on the physical–chemical–sensory properties of Italian Cantuccini biscuits. Foods 2022, 11, 299, https://doi.org/10.3390/foods11030299

[X3] Enhancement of functional and nutritional properties of bread using a mix of natural ingredients from novel varieties of flaxseed and lupine. LWT 91, 48-54 (2018).

Response 6: A number of 29 new reference titles were added, and the following studies were introduced in the introduction section.

Scientific research in the field of functional ingredients in the bakery industry has led to the production of products with varying degrees of replacement of wheat flour, with partially defatted hemp seeds [17], Jerusalem artichoke tubers [18], oats [19], buckwheat [20], green tea [21], Pleurotus Ostreatus fibers 1-3 , 1-6 Beta-glucan [22], various by-products from the fruit and vegetable processing industry, millet [23], sorghum [24], amaranth [25, 26], quinoa [27], sunflower seeds [28], flaxseeds [29], pumpkin seeds [30], lupine [31], chia seeds [32], peas [33], chickpeas [34], lentils [35], chestnuts [36], cricket flour [37], etc., with important effects in terms of increasing the content of fiber, minerals, protein, vitamins, antioxidants, etc.

Laganà et al. [38] have used bergamot Pastazzo flour (obtained after juice extraction) wich is known for it’s high content in antioxidants, and obtained shortbread biscuits. Pastazzo flour (from pressed pulp) is usually used in animal feed or it is discharged. The bergamot Pastazzo flour was used in different percentages (2.5%, 5%, 10% and 15%), and the obtained results showed that antioxidant content increases once the Pastazzo flour addition is increased.

Giuffrè et al. [39] have fortified with olive oil an original recipe of Italian Cantuccini biscuits, using up to 70% extra virgin olive oil instead of 50% margarine and reducing with 20% the addition of cow butter. The aim of the study was was to evaluate the shelf-life and physicochemical properties of biscuits and of the fats contained in original recipe Italian Cantuccini biscuits, and also the sensory properties were evaluated , including colour, appearance, taste, flavour, texture and overall acceptability.

Wandersleben et al. [40] have conducted a study using three types of ingredients: lupine grit flour (AluProt-CGNA®, 60% of protein, d. m.), lupine hulls flour and flaxseed expeller flour and tested the dough rheological properties of different combinations of the studied ingredients with wheat flour and also the consumer's acceptability. The results show that the addition of natural sources of protein and dietary fibre through lupine or flaxseed flour to the wheat bread improved it’s nutritional profile. It was found the appropriate blend to attenuate the effect of foreign ingredients over the bread rheology, which normally interfere with the gluten network reducing the quality of the bread, and also enhance its nutritional value.

Multescu et al. [41] studied the phenolic content, flavonoid content, and the lipid-soluble antioxidant capacity of 14 byproducts (rapeseed meals, grape seed flour, sun flower meals, seabuckthorn flour, etc.) obtained in the vegetable oil industry. Results confirmed that the byproducts analyzed are a valuable source of many biological functional substances having considerable amounts of total phenolic content. The studied byproducts can be used as ingredients for new bakery products in order to improve their nutritional properties and antioxidant quality.

Point 7. Introduction section line 81 and in the whole manuscript: Mironeasa et al… have investigated. It is a group of researchers;

Response 7: The changes were made throughout the manuscript as suggested;

Point 8. Caption of table 1: Table and not table;

Response 8: The mistake has been fixed.

Point 9. Table 1 in the caption you have written defatted, in the table you have written degreased. Be consistent in the whole manuscript, I suggest defatted;

Response 9: The changes were made throughout the manuscript as suggested, thank you;

Point 10. 2 section and in the whole manuscript, tables and figures, when you indicate the temperature, separate the numeric value from the symbol: 40-65 °C and not 40-65°C;

Response 10: All numerical values were separated from the symbols throughout the manuscript;

Point 11. 3 sub-section, line 118 and in the whole manuscript: separate decimal by a dot and not by a comma;

Response 11: Dots were used instead of comma, throughout the manuscript as suggested;

Point 12. 5 sub-section, line 149: what 80 is? Rpm or what? In addition, write -1 as an exponent (if you are meaning … per minute);

Response 12: The mistake was fixed, indeed, it is 80 rpm.

Point 13. 5 sub-section, line 173 and in the whole manuscript, please, use the correct type for beta (β) from the ancient Greek alphabet, the type you have used is from the modern German alphabet and it is not beta;

Response 13: The mistake was fixed and beta (from Greek alphabet) was added;

Point 14. Table 3, separate decimal by a dot and not by a comma;

Response 14: Dots were used instead of comma, throughout the manuscript as suggested;

Point 15. 8 sub-section and in the whole manuscript. When you have indicated the significance you have used different criterion for spacing: p < 0.05 or p<0.05?, In addition p italicized or not?

Response 15: The changes were made, using an italicized p and spacing: p < 0.05;

Point 16. Line 263 and in the whole manuscript, do not insert a dot after g. The dot after g only at the end of a sentence

Response 16: The dot was deleted;

Point 17. Caption of table 5, insert one space after .. day;

Response 17: The space was added after the word „day”;

Point 18. Discussion section (Magnesium), Conclusions section (magnesium). Please be consistent in the whole manuscript for this and for all chemical compounds;

Response 18: All chemical compounds were written without capital letters, unless they are the first word of a sentence;

Point 19. References section. This section has to be arranged in light of the instructions for authors of Foods. For example, the journal name has to be abbreviated (ref 15). The capital letter only for the first type. Please verify in the template;

Response 19: References have been modified according to the template, and the journal names were abbreviated;

Point 20. The references section is not arranged as requited by Foods;

Response 20: References have been modified according to the template;

Point 21. References section, ref 28: ect?

Response 21: The reference no 28 (now 57) was fixed. „_ „symbol was chnaged to double f letter.

Point 22. Please, very important, write in blue color or evidence differently the corrections you will do.

Response 22: All changes were made using track changes as suggested by the editor. Thank you.

Round 2

Reviewer 1 Report

Authors utilize the skills for manuscript improvement. clear for acceptance now

Author Response

Thank you very much for your support!

Reviewer 2 Report

Manuscript Number: foods-1739811, titled:

 Research on the potential use of grape seed flour in the bakery industry

Review 2 – 24 May 2022

Dear Editor of Foods

the argument is interesting and the authors have included the large part of my comments. The References section contains inaccuracies and has to be completed and arranged using the instructions for authors of Foods. I have listed some correction to do in this section but the Authors have to carefully verify line by line, word by word.

I suggest a minor revision.

To the Authors (in detail):

1)     The references section is not arranged as requited by Foods and has to be verified word by word;

2)     References section. This section has to be arranged in light of the instructions for authors of Foods. For example, the journal name has to be abbreviated and the capital letter only for the first type. Please verify in the template;

3)     References section, sometime you have written the title in small letters (ref 12, 36) and sometime in capital letter (ref 11), please, be consistent in the whole section;

4)     References section, ref 16: for example, the abbreviation of Grasas y Aceites is Grasas Aceites and not Gras. Y Ace;

5)     The abbreviations of journals are codified and are well known by all Authors. In this section almost all abbreviations are uncorrected. Please digit: GOOGLE > journal title abbreviations > choose the first letter of each journal (for example g for Grasas y Aceites> find the correct abbreviation> replace the one in your manuscript;

6)     You can find the abbreviation also directly on Google, digit: abbreviation and the title of the journal. In many cases you will have the correct abbreviation;

7)     Ref 17 and in the whole section: of … is not included for abbreviations;

8)     Ref 11, verify this abbreviation. If it does not exist, write the complete name;

9)     Ref 12 and in the whole section: the abbreviation of the Botanist has not to be italicized. Please, use the international binomial nomenclature: L. and not L. italicized;

10) Ref 21 and in the whole section, verify if & is required by the guidelines of Foods;

11) Please, very important, write in blue color or evidence differently the corrections you will do.

I suggest a minor revision

Regards.

Author Response

Point 1. The references section is not arranged as requited by Foods and has to be verified word by word;

Response 1: The reference list was verified and arranged as required by Foods. Thank you.

Point 2. References section. This section has to be arranged in light of the instructions for authors of Foods. For example, the journal name has to be abbreviated and the capital letter only for the first type. Please verify in the template;

Response 2: The abbreviations were corrected and capital letter was used only for the first word of each title.

Point 3. References section, sometime you have written the title in small letters (ref 12, 36) and sometime in capital letter (ref 11), please, be consistent in the whole section;

Response 3: The titles were all changed to small letters, except for the first word of each title.

Point 4. References section, ref 16: for example, the abbreviation of Grasas y Aceites is Grasas Aceites and not Gras. Y Ace;

Response 4: The change was done as suggested.

Point 5. The abbreviations of journals are codified and are well known by all Authors. In this section almost all abbreviations are uncorrected. Please digit: GOOGLE > journal title abbreviations > choose the first letter of each journal (for example g for Grasas y Aceites> find the correct abbreviation> replace the one in your manuscript;

Response 5: The changes were done as suggested. Thank you.

Point 6. You can find the abbreviation also directly on Google, digit: abbreviation and the title of the journal. In many cases you will have the correct abbreviation;

Response 6: The search for the correct abbreviation of each journal was made on Google, and ISO 4 abbreviations were selected.

Point 7. Ref 17 and in the whole section: of … is not included for abbreviations;

Response 7: The changes were done as suggested.

Point 8. Ref 11, verify this abbreviation. If it does not exist, write the complete name;

Response 8: The journal has no abbreviation form so the complete name was used.

Point 9. Ref 12 and in the whole section: the abbreviation of the Botanist has not to be italicized. Please, use the international binomial nomenclature: L. and not L. italicized;

Response 9: Changes were done as suggested.

Point 10. Ref 21 and in the whole section, verify if & is required by the guidelines of Foods;

Response 10: The “&” symbol was deleted.

Point 11. Please, very important, write in blue color or evidence differently the corrections you will do.

Response 11: All changes were done with track changes. Thank you.